

# Response of Arctic ozone to sudden stratospheric warmings

Alvaro de la Cámara[1,2], Marta Abalos[1], Peter Hitchcock[3], Natalia Calvo[1], and Rolando R. Garcia[4]

[1]Dept. Física de la Tierra y Astrofísica, Universidad Complutense de Madrid (UCM), Spain.
[2]Instituto de Geociencias (IGEO), CSIC-UCM, Madrid, Spain.
[3]Laboratoire de Météorologie Dynamique /IPSL, Ecole Polytechnique, Palaiseau, France.
[4]National Center for Atmospheric Research, Boulder CO, United States.

**Correspondence:** Alvaro de la Cámara (acamarai@ucm.es)

**Abstract.** Sudden stratospheric warmings (SSW) are the main source of intra-seasonal and interannual variability in the extratropical stratosphere. The profound alterations to the stratospheric circulation that accompany such events produce rapid changes in the atmospheric composition. The goal of this study is to deepen our understanding of the dynamics that control changes of Arctic ozone during the life cycle of SSWs, providing a quantitative analysis of advective transport and mixing.

We use output from four ensemble members (60 years each) of the Whole Atmospheric Community Climate Model version 4 performed for the Chemistry Climate Model Initiative, and also use reanalysis and satellite data for validation purposes. The composite evolution of ozone displays positive mixing ratio anomalies up to 0.5 – 0.6 ppmv above 550 K (∼50 hPa) around the central warming date and negative anomalies below (-0.2 to -0.3 ppmv), consistently in observations, reanalysis and model. Our analysis shows a clear temporal offset between ozone eddy transport and diffusive ozone fluxes. The initial changes in

ozone are mainly driven by isentropic eddy fluxes linked to enhanced wave drag responsible for the SSW. The recovery of climatological values in the aftermath of SSWs is slower in the lower than in the upper stratosphere, and is driven by the competing effects of cross-isentropic motions (which work towards the recovery) and isentropic irreversible mixing (which delays the recovery). These features are enhanced in strength and duration during sufficiently deep SSWs, particularly those followed by Polar-night Jet Oscillation (PJO) events. It is found that SSW-induced ozone concentration anomalies below 600 K (∼40

hPa), as well as total column estimates, persist around one month longer in PJO than in non-PJO warmings.

## 1 Introduction

Understanding the impact of dynamical processes such as sudden stratospheric warmings (SSWs) (e.g., Butler et al., 2015) on Arctic ozone is key to interpreting the observed interannual variability and better quantify polar ozone evolution (WMO, 2014).

The stratospheric circulation distributes ozone far from its photochemical production region in the tropics (e.g., Solomon et al., 1986; Hauchecorne et al., 2002). The global distribution of ozone is largely controlled by a balance between advection by the stratospheric overturning circulation, rapid isentropic stirring and mixing that follows planetary Rossby wave breaking, and



chemical sources and sinks (e.g., Hartmann and Garcia, 1979; Garcia and Solomon, 1983; PLUMB, 2002). The extreme and transient nature of the dynamical forcing that triggers SSWs alters this balance. Driven by an abrupt growth of wave activity (e.g., Matsuno, 1971; McIntyre, 1982; Birner and Albers, 2017; de la Cámara et al., 2017), SSWs produce global changes in the middle atmospheric circulation that affect polar temperatures –and thus ozone depletion (Newman et al., 2001)–, and

impact tracer transport and mixing (Randel et al., 2002). In a recent study, de la Cámara et al. (2018) used reanalysis data and model output from the Whole Atmosphere Community Climate Model (WACCM) to provide a composite view of the changes in transport and mixing properties of the flow during the life cycle of SSWs. They found that after the onset of SSWs, the residual circulation remains weak as a result of suppressed wave driving, but enhanced mixing nonetheless persists in the lower stratosphere for over two months. This study also showed a clear temporal offset between wave forcing and mixing;

zonal-mean eddy fluxes of potential vorticity (PV) decay short after the SSW onset, while diffusive PV fluxes (in equivalent latitudes) remain active several weeks after. de la Cámara et al. (2018) found these anomalies in transport and mixing stronger and more persistent for those warming events that occur during a Polar-night Jet Oscillation (PJO) event (Kodera et al., 2000; Hitchcock et al., 2013a). Note that the notion of PJO events in the present paper is similar to that in Hitchcock et al. (2013a) and de la Cámara et al. (2018) in the sense that PJO events are associated explicitly with SSWs (i.e. sufficiently deep SSWs).

This differs from the perspective of Kodera et al. (2000) who saw the PJO as a low-frequency, stratospheric mode of variability that sometimes phase-locks with SSWs.

   Case studies for several Arctic winters based on the combined use of observations and Lagrangian transport models highlight the wide range of inter-event variability and the sensitivity of polar chemical processing to the different dynamical conditions. Manney et al. (2003) used a Lagrangian transport model and ozone data from the Aura Microwave Limb Sounder (MLS)

to estimate chemical ozone depletion in the polar vortex for the 1991/92 through 1997/98 boreal winters. They found large interannual variability in the timing and spatial patterns of ozone depletion due to variability in the position of the vortex and dynamical processes. Konopka et al. (2007) combined satellite observations of ozone from POAM (Polar Ozone and Aerosol Measurement III) and MIPAS (Michelson Interferometer for Passive Atmospheric Sounding), with simulations of the Chemical Lagrangian Model of the Stratosphere (CLaMS) to study the 2002/03 Arctic winter. They found that strong wave

events associated with the 2003 SSW may have increased tracer transport and enhanced chemical ozone destruction in the polar vortex and its surroundings. The SSW on January 2009 was one of the strongest on record (e.g., Ayarzagüena et al., 2011; Albers and Birner, 2014). Both Manney et al. (2009) and Tao et al. (2015) highlight the enhanced isentropic mixing of trace gases across the vortex edge after the onset of the 2009 SSW, in agreement with the composite analysis of de la Cámara et al. (2018). Another example of the impact of a sudden warming on Arctic ozone is the 2015/2016 Northern Hemisphere winter,

which was one of the coldest in the polar stratosphere in recent years. Intense ozone loss developed in February 2016 favored by the low Arctic temperatures, but was abruptly terminated by a sudden warming in early March that became one of the earliest final warmings on record (Khosrawi et al., 2017; Manney and Lawrence, 2016). Strahan et al. (2016) used reanalysis meteorological fields to integrate the Global Modeling Initiative (GMI) chemistry transport model for the 2005-2015 boreal winters, and estimated that winters with SSWs before mid-February have about 1/3 the depletion of winters without SSWs.

However, the cold, undisturbed vortex conditions of December 2012, and the subsequent vortex split of early January that



produced unusually long-lasting offspring vortices subject to high sunlight exposure, led to exceptionally high ozone depletion in January 2013 (Manney et al., 2015).

The present study aims to provide a quantitative evaluation of the changes in Arctic ozone induced during SSW events, using the Whole Atmosphere Community Climate Model (WACCM). The use of output from a free-running state-of-the-art

chemistry-climate model facilitates the evaluation of the separate contributions of transport/mixing and chemical processes to the ozone variations during the dynamically active boreal winter stratosphere, which is always a challenge for observational studies (Livesey et al., 2015). The 240 years of WACCM output will also provide the statistical robustness that the relatively short observational record lacks. We will show that the ozone field in WACCM shares many features with observations of the Microwave Limb Sounder (MLS) on Aura satellite, and with reanalysis. The evaluation of the different terms of the zonal-

mean ozone continuity equation in geometric latitude on isentropic levels, combined with the analysis of irreversible mixing diagnostics in equivalent latitude, will show that ozone anomalies during SSWs are mainly controlled by dynamical processes in the mid-to-lower stratosphere. In addition, sudden warmings that occur during a PJO event have stronger dynamically-induced ozone anomalies that persists around one month longer than warmings without a PJO event.

The reminder of the paper is organized as follows. Section 2 describes the model runs, the observational data and reanalysis

used, and the diagnostics employed. Section 3 presents and discusses the results, and section 4 gives the main conclusions.

## 2 Data and methods

### 2.1 Model output and data

WACCM is a global chemistry climate model developed at the National Center for Atmospheric Research (NCAR), and can be used as the atmospheric module of the Community Earth System Model (CESM). The version used in this study, version

4 (Marsh et al., 2013), has a horizontal resolution of 2.5°x1.9° longitude–latitude, and 66 levels in the vertical with the top at about 140 km altitude. A few updates from (Marsh et al., 2013) include a new chemistry module with revised heterogeneous chemistry (Wegner et al., 2013; Solomon et al., 2015), and changes to the orographic gravity wave parameterizations that significantly reduce the Antarctic cold pole bias (Garcia et al., 2017). We use daily averaged fields from 4 members of an ensemble of 60-year climate simulations (a total of 240 years, each ensemble member only differs in slightly different initial

conditions of the atmospheric state) originally designed for the Chemistry Climate Model Initiative (CCMI) (Eyring et al., 2013; Morgenstern et al., 2017). These runs are forced with observed sea surface temperatures and external forcings for the period 1955-2014 (i.e. the CCMI REF-C1 configuration), and the quasi-biennial oscillation is nudged by relaxing the stratospheric tropical zonal winds towards observations (Marsh et al., 2013).

Two observational datasets and one reanalysis product are used for validation purposes. Ozone mixing ratio from the Strato-

spheric Water and Ozone Satellite Homogenized (SWOOSH) data set for the period 1984-2017 (Davis et al., 2016) is used for a seasonal cycle comparison with the model. This monthly-mean merged ozone product combines observations from a set of satellite instruments (SAGE-II/III, UARS HALOE, and UARS and Aura MLS) after a homogenization process to account for inter-satellite biases and to minimize artificial jumps in the record (see Davis et al., 2016, for more information). For compar-





isons of the evolution of Arctic ozone during SSWs, we use daily averaged zonal-mean ozone mixing ratio from Aura MLSv3, which covers the period September 2004 – July 2012 on a 7.5° latitude grid, and daily output of ozone mixing ratio from the European Interim Reanalysis (ERAI), produced by the European Center for Medium-Range Weather Forecasts (Dee et al., 2011), for the period 1979-2012 on a 1°x1° longitude–latitude grid. Dragani (2011) determined that the observation minus
analysis residuals of ozone in ERAI are typically within ±5% in the region of the ozone mixing ratio maximum at 10 hPa and above, but larger up to around 20% in the lower stratosphere.

## 2.2    Diagnostics of ozone transport

We will use daily-mean WACCM output to evaluate the continuity equation of zonal-mean ozone concentration ($\chi$) on isentropic levels (e.g., Andrews et al., 1987):

$$\partial_t \bar{O}_3 = \bar{S}^* - \underbrace{a^{-1}\bar{v}^*\partial_\phi \bar{O}_3 - \bar{Q}^*\partial_\theta \bar{O}_3}_{mean\ advection} + \underbrace{\bar{\sigma}_\theta^{-1}\left[(a\cos\phi)^{-1}\partial_\phi(M_\phi\cos\phi) + \partial_\theta M_\theta\right]}_{eddy\ transport} - \bar{\sigma}_\theta^{-1}\partial_t \overline{\sigma_\theta' O_3'}, \qquad (1)$$

where $O_3$ denotes ozone mixing ratio, $S$ is the net ozone tendency due to chemistry (chemical production minus loss), $(v,Q)$ are the meridional and cross-isentropic velocities ($Q$ is the diabatic heating rate), $\sigma_\theta \equiv -g^{-1}\partial_\theta p$ is the isentropic density, $a$ is the Earth radius, $\phi$ is latitude, $\theta$ is potential temperature, and $t$ is time. The vector $\boldsymbol{M} = (0, M_\phi, M_\theta) = (0, -\overline{(\sigma_\theta v)'O_3'}, -\overline{(\sigma_\theta Q)'O_3'})$ is the eddy flux term, whose divergence can be interpreted as an eddy transport term. Overbars denote the zonal mean, primes
departures from it, and stars denote mass-weighted, zonally-averaged variables ($\bar{X}^* = \bar{\sigma}_\theta^{-1}\overline{\sigma_\theta X}$). Note that the second and third terms in the right-hand side of Eq. 1 represent advection (isentropic and cross-isentropic, respectively) by the zonal-mean overturning circulation. Also, the last term is usually quite small and will not be shown, but it has been taken into account to compute the balances.

     The eddy transport term is frequently used as an estimate of the two-way mixing effect of Rossby wave breaking on tracer
concentrations, to distinguish it from the mean advective transport by the residual circulation (e.g., Abalos et al., 2013). However this eddy transport term, computed as the divergence of the eddy tracer flux, does not completely separate the irreversible two-way mixing and it can include a component of reversible transport (see de la Cámara et al., 2018). Furthermore, there can be eddy transport of chemical species that is irreversible in the absence of wave breaking. This can occur when the waves are dissipated thermally, or when the chemical lifetime of a species changes along the wave trajectory (i.e., chemical eddy transport
(Hartmann and Garcia, 1979; Garcia and Hartmann, 1980)). To explore in more detail the role of *irreversible mixing* in ozone tendencies during SSWs, we evaluate the normalized equivalent length squared (hereafter simply *equivalent length*) for ozone (Haynes and Shuckburgh, 2000):

$$\Lambda_{eq}^{O_3}(\phi_e, t) \equiv a^2 \left\langle |\boldsymbol{\nabla}_{\theta\ const}\ O_3|^2 \right\rangle (\partial_{\phi_e} O_{3,e})^{-2}, \qquad (2)$$

where $\langle \cdot \rangle$ represents the area-average between consecutive tracer contours, and $O_{3,e}$ is ozone mixing ratio in equivalent lat-
itude $\phi_e$ (EqL) coordinates (Allen and Nakamura, 2003). This coordinate system assigns the area $A$ enclosed by a given tracer contour to a circle of latitude (i.e., the equivalent latitude) that is the boundary of the polar cap with the same area,



$A = 2\pi a^2(1 - \sin\phi_e)$. $\Lambda_{eq}$ is proportional to the effective diffusivity $\kappa_{eff}$ (Nakamura, 1996) ($\Lambda_{eq} = \kappa_{eff}/\kappa$), with $\kappa$ a constant diffusion parameter that depends on the model's spatial resolution and the hyperdiffusivity scheme employed. The equivalent length (or the effective diffusivity) quantifies the changes in microscale diffusion due to the irreversible elongation of tracer contours mainly caused by large-scale Rossby wave breaking and subsequent stirring. In EqL coordinates, the continuity

equation for ozone will therefore be given by:

$$\partial_t \chi_e = -(a\cos\phi_e)^{-1}\partial_{\phi_e}\left(F_d^{O3}\cos\phi_e\right) + \text{ (diabatic and chemical terms)} \qquad (3)$$

where $F_d^{O3} = -a^{-1}\kappa_{eff}\partial_{\phi_e}\chi_e$ is the horizontal diffusive flux of ozone in EqL. Estimating the explicit value of $F_d^{O3}$ is challenging since the constant diffusion parameter $\kappa$ of the model, which enters in the definition of $\kappa_{eff}$, is unknown. However, we will employ the equivalent length $\Lambda_{eq}^{O_3}$ instead of the effective diffusivity $\kappa_{eff}$ to compute $F_d^{O3}$ as they basically contain the same

information. This will not affect our results since we are interested in the anomalies of this diagnostic during SSWs (and not in its absolute value). Also, note that there is no horizontal (isentropic) advection term (either by the mean flow or by the eddies) in Eq. 3 since it is embedded in the tracer-based coordinate system. The only horizontal process involved in the evolution of ozone in EqL is the first term in the right-hand side, which represents the mixing-induced ozone tendency.

## 2.3   Methodology

Sudden stratospheric warmings are identified in ERAI and WACCM applying the widely-used criterion of Charlton and Polvani (2007). The day when the zonal-mean zonal wind at 60°N and 10 hPa turns negative is set as the central warming date, provided that it occurs between November and March (i.e., mid-winter warmings), the separation from the previous central date is longer than 20 days, and the wind returns to positive values for at least 10 consecutive days before April 30th. In the 34-year period of ERAI we identify 23 SSWs ($0.68 \, \mathrm{yr}^{-1}$), while in the 240 years of WACCM simulations we have 152 SSWs ($0.63 \, \mathrm{yr}^{-1}$). In

the MLS period (Sep 2004 - July 2012), we have 6 events.

We classify SSWs depending on whether or not they occur during PJO events. These events are identified following the procedure of Hitchcock et al. (2013a). Briefly, the PJO classification is done in terms of the first two Empirical Orthogonal Functions (EOFs) of daily-mean polar-cap-averaged (70°-90°N) temperatures over the middle-atmospheric column, which present both a vertical dipole structure (Kodera et al., 2000). A PJO event is identified when the temperature anomaly (as projected

onto these two EOFs) maximizes at a height of approximately 60 hPa, so long as it is sufficiently strong (see Hitchcock et al., 2013a, for further details). Consequently, SSWs that occur during PJO events will have a strong signal in the lower stratosphere, but note that the identification criterion does not explicitly consider the persistence of the anomalies. We find that 70 SSWs occur during PJO events (hereafter PJO-SSW) in WACCM, while 82 are not linked to PJO events (hereafter nPJO-SSW).

The methodology followed consists on constructing composites of the fields as a function of latitude or altitude, centered on

the SSW central date. The daily anomalies are calculated as the difference between the daily value and the daily climatological average (smoothed with a 10-day running mean). The statistical significance is assessed applying a two-tailed Student t-test to compare the composite mean of SSWs and the climatology. We use N-1 degrees of freedom, N being the number of SSWs





included in the composite, and a confidence level of 99% (i.e., $\alpha = 0.01$). Each SSW event has been assumed to be independent to estimate the degrees of freedom.

## 3   Results

### 3.1   Annual cycle of ozone in observations, reanalysis and WACCM

We first compare the Northern Hemisphere seasonal cycle of ozone in SWOOSH, ERAI and WACCM. Figure 1 shows the seasonal evolution of zonal-mean ozone mixing ratio at 10, 70 and 100 hPa (or the nearest levels available) for the three data sources. As expected, in all datasets the latitudinal gradients have opposite signs in the lower and middle stratosphere (e.g., Brasseur and chemist) Solomon, 2005): Ozone mixing ratio over the Arctic is smaller than in mid-latitudes at 10 hPa, and the opposite is true at 70 and 100 hPa. Also, at 70 and 100 hPa the seasonal cycle is characterized by maximum values in
winter and minimum in summer (consistent with the overturning circulation seasonality), while at 10 hPa the minimum values occur in autumn over the polar cap. WACCM and ERAI present very similar values at the three levels shown, and there is good agreement with SWOOSH at 10 hPa. In the lower stratosphere (70 and 100 hPa) WACCM and ERAI agree well with observations, although both the model and reanalysis present mixing ratios around 10% larger than SWOOSH in winter over the Arctic, which matches the findings of Dragani (2011).

Figure 2 shows the contribution of each term in Eq. (1) to the simulated seasonal cycle of the ozone budget in WACCM, averaged over 70°-90°N on the isentropic levels of 850 K (∼10 hPa), 500 K (∼60 hPa) and 400 K (∼100 hPa). Note the transient eddy term (last term in Eq. 1) is usually very small and not shown here. At 850 K isentropic eddy transport and net chemical loss nearly balance each other, particularly from February to May, and vertical advection makes a small contribution in autumn and winter. As a result, the tendency is a small residual relative to these two competing effects. The polar middle
stratosphere constitutes the transition layer above which ozone is chemically controlled, and below which it is dynamically controlled (e.g., Brasseur and chemist) Solomon, 2005). This is evident in Figs. 2b and 2c, where the ozone budget terms are displayed at 500 and 400 K. At 500 K, chemical destruction is still relevant in spring and summer, but the shape of the ozone seasonal cycle is mainly determined by the seasonally varying cross-isentropic advection and isentropic eddy transport (although the chemical sink in late spring and early summer delays the ozone minimum to mid-summer). Downward motion in
winter increases ozone over the pole, while isentropic eddy transport works against this, smoothing out the ozone meridional gradients. At 400 K, the chemical term is practically irrelevant, and the seasonal budget of ozone is completely controlled by the competing effects of cross-isentropic advection and isentropic eddy transport.

The good agreement in the ozone seasonal cycle between WACCM and observations, as well as the reproduction of well-known features in the ozone budget, allows us to explore in the next subsections the driving mechanisms of ozone changes
during the lifetime of SSWs using WACCM.



### 3.2 Changes in polar ozone during SSWs

Figures 3a, 3b and 3c show the composite anomalies of ozone in MLS, ERAI and WACCM, respectively, averaged over the Arctic (70°-90°N) as a function of time lag with respect to the SSW central date. The three panels show very similar behavior despite the variety of datasets and years covered (note that the MLS composite is based on only 6 events). Arctic ozone mixing
ratio is enhanced (0.5-0.6 ppmv) at levels where ozone decreases poleward ($> 550$ K), and reduced (0.2-0.3 ppmv) at levels where it increases poleward ($< 550$ K). MLS and ERAI show a sharper growth of the anomalies from lags -5 to 0 d (Figs. 3a and 3b), while in WACCM the growth is more gradual over the course of the 10 days preceding the central date (Fig. 3c). At positive lags there is a slow 'descent' of positive ozone anomalies in the mid-stratosphere, and ozone returns to pre-warming values faster in the upper than in the lower stratosphere.

We can now take full advantage of WACCM meterological fields and investigate the driving mechanisms of these anomalies during SSWs by evaluating the different terms in the zonal-mean ozone budget equation (Eq. 1). Figure 4 shows the anomalies of the most relevant terms of Eq. 1, including the ozone tendency (Fig. 4a), the isentropic and cross isentropic mean advection (Figs. 4c and 4d, respectively), the isentropic eddy transport (Fig. 4e), and the chemical production minus loss (Fig. 4f). The cross-isentropic eddy transport and eddy transient terms (the last two terms in Eq. 1) are very small and will not be shown. Note
that instead of showing the residual, Fig. 4b displays the ozone tendency ($\partial_t \bar{O}_3|_b$) that results from the sum of all the terms in the right-hand side of Eq. 1, and it can be compared with the direct calculation of the ozone tendency in Fig. 4a. There is a relative good agreement between the direct and "net" calculations of the ozone tendency below $\sim 700$–$800$ K (Figs. 4a and 4b). However, some discrepancies appear at $\theta > 800$ K, which are likely due to uncertainties in the calculation of the eddy transport term (note that periods of large discrepancy between $\partial_t \bar{O}_3$ and $\partial_t \bar{O}_3|_b$ at $\theta > 800$ K, such as at lags 45–60 d and 75 d, coincide
with periods of very large anomalies of isentropic eddy transport at those levels in Fig. 4e). Another source of discrepancy is that we do not include in Eq. 1 the effects of vertical diffusion due to the gravity wave parameterization in WACCM, which presumably are non-negligible in the mid-to-upper stratosphere.

Over the two weeks prior to the central date (lags -15–0 d), the isentropic eddy transport leads off the ozone changes (Fig. 4e). This indicates that the initial increase of ozone mixing ratio at negative lags above and decrease below $\sim 550$ K
(Figs. 3c and 4a) is primarily a consequence of the growth of planetary waves in the stratosphere that ultimately triggers the SSW. Other terms of Eq. 1 make a relatively smaller albeit significant contribution at this early stage, such as a growing cross-isentropic advection that increases ozone below 900 K and decreases ozone above (Fig. 4d), negative anomalies of isentropic mean advection at levels higher than 500 K, and a large negative chemical tendencies above 700 K (Fig. 4f) that tends to restore photochemical equilibrium in response to the dynamically-induced ozone anomalies. In the aftermath of the warming (positive
lags), the anomalies of cross-isentropic advection present a downward-progressing structure (Fig. 4d) that leads to a gradual return to climatological values of ozone below 500 K. Above 500 K in the mid and upper stratosphere, where wave activity is suppressed in the aftermath of the warming (e.g., Limpasuvan et al., 2004; Hitchcock and Shepherd, 2013; de la Cámara et al., 2017), reduced isentropic eddy transport and cross-isentropic advection allows ozone mixing ratio to recover at a much faster rate (Fig. 4e), while chemical tendencies partially counteract these effects.



Extracting the effects of irreversible mixing from the Eulerian-mean eddy transport term is a challenging task, and no effort will be done here. Instead, we use the equation for the evolution of ozone in the equivalent latitude EqL framework (Eq. 3), where the only isentropic process that modifies ozone is irreversible mixing (see section 2.2). Figure 3d shows the evolution of ozone anomalies averaged over EqL $\phi_e = 70° - 90°$N. The structure of ozone anomalies in EqL is overall similar to that in geographical coordinates (Fig. 3c), but there are details that provide a somewhat different perspective. The level of maximum positive anomalies in EqL appears to be located at altitudes higher than 1100 K (while in geographical coordinates it is located at 700–800 K). But the most significant difference is that the initial changes in ozone appear around one week later in EqL than those over the geographical polar cap at all levels (compare Figs. 3c and 3d). It should be noted that the average over $\phi_e = 70°$-$90°$N in EqL encompasses the interior of the polar vortex, especially at negative lags. Therefore, the initial ozone increase over the Arctic (in geographical coordinates) above 600 K that starts at lag -15 (Fig. 3c) does not happen inside the vortex; otherwise it would have been captured in EqL coordinates. Additionally, we conclude now that those changes should be a consequence of reversible isentropic eddy transport in Fig. 4e (as opposed to irreversible mixing), since ozone in EqL (which can only be changed by non-conservative processes, see Eq. 3) does not present those changes.

To explore this feature in more detail, the left column of Fig. 5 shows the evolution of the anomalies of ozone tendency in EqL (Fig. 5a) and mixing-induced ozone tendency (first term in the right-hand side of Eq. 3) (Fig. 5d) averaged over the EqL $\phi_e = 70° - 90°$N during SSWs. Note that the central and right panels in Fig. 5 will be discussed in section 3.3. The anomalies of the mixing-induced tendency term have been normalized by the standard deviation at each isentrope since we cannot compute the absolute value of the diffusive flux of ozone $F_d^{O3}$ (see section 2.2). A comparison between Figs. 5a and 5d reveals that the initial changes in ozone in EqL share timing with enhanced irreversible mixing, which tends to reduce ozone below $\sim$600 K and increase ozone above (Fig. 5d). The ozone-induced mixing anomalies persist well after the onset of SSWs, up to lag 30 d in the mid and upper stratosphere and up to lag 45 d in the lower stratosphere.

Consistent to what was mentioned at the end of the previous paragraph, there is practically no sign of enhanced mixing at negative lags, confirming the reversible nature of the Eulerian-mean isentropic eddy transport increase at negative lags in Fig. 4e. The timing and duration of the mixing-induced ozone tendencies are dominated by the behavior of the anomalies of the equivalent length $\Lambda_{eq}^{O3}$, which are shown in the left column of Fig. 6 at the 850, 450 and 400-K isentropes (the central and right panels in Fig. 6 will be discussed in section 3.3). The anomalies of $\Lambda_{eq}^{O3}$ during SSWs have a similar latitudinal structure and evolution to $\Lambda_{eq}$ computed from the PV field at these levels (de la Cámara et al., 2018), which emphasizes that the evolution of ozone is dominated by the dynamics. Positive anomalies of $\Lambda_{eq}^{O3}$ (enhanced mixing properties) start at 850 K and lag -10 in the mid-latitudes, migrating poleward at positive lags lasting until lag 30, and being replaced by a period of weak mixing. As we move down to lower altitudes the positive anomalies of $\Lambda_{eq}^{O3}$ appear increasingly delayed, and persist for longer than 2 months at 450-400 K.

de la Cámara et al. (2018) showed that the response of irreversible mixing to wave breaking during SSWs is not instantaneous, but extends over several weeks (as long as 2 months in the lower stratosphere) after the large-scale wave forcing has decayed. This behavior is reproduced in the comparison of the zonal-mean isentropic eddy transport of ozone $((a\cos\phi)^{-1}\partial_\phi(M_\phi\cos\phi))$ and the mixing-induced ozone tendency in EqL $(-(a\cos\phi_e)^{-1}\partial_{\phi_e}(F_d^{O3}\cos\phi_e))$ in Figs. 4e and 5d, respectively. On one hand,





the Eulerian-mean eddy transport term increases ozone above and decreases ozone below ~600 K (i.e., smooths out the horizontal gradients) at negative lags, and then the anomalies reverse sign following tightly the behavior of the wave forcing during SSWs (e.g., Limpasuvan et al., 2004). On the other hand, the ozone gradient-smoothing effect of enhanced irreversible mixing persists long after the wave forcing (and the Eulerian-mean eddy transport) have declined. Part of this temporal offset may

be explained as follows. The wave forcing (e.g. EP flux divergence from a TEM perspective or meridional eddy PV transport from an Eulerian perspective) distorts PV contours in geometric coordinates and fluxes PV (and ozone) across latitude circles (zonal-mean isentropic eddy transport). This is fast, occurs in the week or two prior to the central dates of the composite, and is reversible (e.g. if the wave packet propagates through and the contours return to zonal). At this point the air within the vortex in EqL has not changed so the lack of ozone anomalies in EqL prior to the central date is consistent (Fig. 3d). After the planetary

waves break, the ozone (and PV) contours remain perturbed with smaller-scale motions, giving rise to slow irreversible mixing. Indeed, the role of non-conservative processes such as mixing in the aftermath of SSWs is to damp wave activity and delay the recovery of the vortex, particularly in the lower stratosphere (Lubis et al., 2018; de la Cámara et al., 2018).

### 3.3 Modulation by PJO events

Recent studies have shown that SSWs that occur during a PJO event (PJO-SSW) experience larger alterations in circulation

and temperature than those warmings that occur without a PJO event (nPJO-SSW), particularly in the recovery phase (e.g., Hitchcock and Shepherd, 2013). Figure 7 shows that PJO and nPJO warmings also have different signatures in polar ozone. The vertical structure and evolution of the composite anomalies in PJO and nPJO events, both in polar cap and equivalent latitude averages (top and bottom panels, respectively, in Fig. 7), are similar to those for all the events (Figs. 3c and 3d). However, the magnitude of the anomalies is larger (below 500 K ozone anomalies are twice as large), and their persistence in the aftermath

much longer in PJO than in nPJO SSWs.

The evolution of the different terms in the zonal-mean balance equation (Eq. 1) averaged over the polar cap ($70° - 90°$N) is shown in Fig. 8 at the 850 and 450 K isentropes. We focus first on the days prior to the onset of the SSW, i.e. at negative lags. The tendencies of ozone in this period, positive at 850 K and negative at 450K, are dominated by the isentropic eddy transport term. At 850 K the anomalies of isentropic eddy transport are shorter-lived but with higher peak values in nPJO

than PJO events. At 450 K, the eddy transport anomalies are quite similar in PJO and nPJO events, but the vertical advection starts to build up more strongly around lag -10 d for PJO than for nPJO warmings. We focus next on the aftermath of the SSWs, where the disparate evolution of ozone between PJO and nPJO events has several contributors depending on the vertical level. In the mid-stratosphere at 850 K (Figs. 8a and 8b), the isentropic eddy transport term goes negative more abruptly in the first days after the central date in PJO than in nPJO events, transporting more ozone out of the polar cap. Also, the strong

suppression of wave driving in the aftermath of PJO-SSWs in the mid-stratosphere leads to a super-recovery of very cold polar temperatures (e.g., Hitchcock and Shepherd, 2013). The subsequent stronger positive anomalies of diabatic heating in PJO than in nPJO events (e.g., Hitchcock et al., 2013b; de la Cámara et al., 2018) produce larger negative vertical advection of ozone in the former than in the latter that lasts until lag 60 d (Figs. 8a and 8b). These stronger dynamical tendencies result in larger negative ozone anomalies in the 700–900 K layer starting at lag 40 d in PJO than in nPJO warmings (compare Figs. 7a

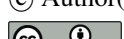



and 7b), which in turn induces stronger positive chemical tendencies trying to restore ozone to its photochemical equilibrium (Figs. 8a and 8b). In the lower stratosphere at 450 K (Figs. 8c and 8d), there are no significant differences between PJO and nPJO events in terms of intensity and timescale of the anomalies prior to lag 0 d. At positive lags, the eddy transport anomalies fluctuate around zero, and the main difference between PJO and nPJO events is the larger contribution (around 3 times as large)

of vertical advection to the recovery of ozone values in PJO than in nPJO events.

Figures 5e and 5f show the mixing-induced tendencies of ozone in EqL coordinates for PJO and nPJO SSWs. Consistently with the stronger ozone anomalies during the former than during the latter, the gradient-smoothing effect of mixing is stronger in PJO than in nPJO events. The central and right panels of Fig. 6 shows that PJO-SSWs produce larger and longer lasting changes in equivalent length of ozone than nPJO events. Again, the differences between PJO and nPJO are more pronounced

in the lower stratosphere: Positive $\Lambda_{eq}^{O3}$ anomalies (i.e., enhanced ozone mixing properties) are three times as large and last over 30 days longer after PJO-SSW than after nPJO-SSWs.

The impact of PJO and nPJO sudden warmings on ozone concentrations agrees well with what is expected from the differentiated responses in the advective overturning circulation and irreversible mixing identified in ERAI and WACCM by de la Cámara et al. (2018) for these two types of warmings. Particularly in the lower stratosphere, they found that the enhanced

mixing and the anomalies of the vertical component of the overturning circulation were twice as strong, and lasted one month longer, in PJO than in nPJO warmings. Hitchcock et al. (2013b) presented evidence indicating that this longer duration of the effects of PJO over nPJO warmings is due to the long radiative timescales in the lower stratosphere (e.g., Dickinson, 1973). PJO-SSW events are characterized by a deeper and stronger penetration of the warming into the lower stratosphere, where radiative relaxation time scales are very slow (Hitchcock et al., 2013b). Along with enhanced, long-lasting diffusive flux of PV

(de la Cámara et al., 2018), they both work to ensure anomalous circulation and temperature conditions for longer times after the onset of the event, delaying the recovery of the vortex.

### 3.4 Response in total ozone column

Several studies have found that interannual variations in winter extratropical total ozone column (TOC) are well correlated with variations of planetary wave activity in the lower stratosphere (e.g., Kinnersley and Tung, 1998; Fusco and Salby, 1999;

Randel et al., 2002). Planetary wave activity affects both the mean advection and mixing of ozone, so those correlations are a simple and useful way of isolating the contribution of dynamics to interannual variations of TOC. There were also early indications that SSWs are followed by large increases in polar TOC after SSWs (e.g. London, 1963).

To complement the analysis of composite changes of ozone based on ozone mixing ratios, we have calculated the resulting zonal-mean anomalies of total ozone column (TOC) during SSWs using MLS, ERAI and WACCM (Fig. 9, top row). The three

composites in the top row of Fig. 9 show a significant increase in TOC north of 45°N with a maximum larger than 25 DU in MLS, and 47 DU in ERAI and WACCM starting a few days before the SSW central date. A TOC reduction larger than 2 DU in MLS, and 3.6 DU in ERAI and WACCM appears south of 45°N; in MLS the reduction of TOC is confined to sub-tropical latitudes. The change of sign of TOC anomalies at ∼45°N is approximately coincident with the climatological position of the maximum latitudinal gradient in WACCM (not shown) and in observations (Garane et al., 2018). As happens with mixing





ratio anomalies, the changes in TOC are present over 40-50 days (Fig. 9) after the SSW onset. In addition, the bottom row of Fig. 9 shows the corresponding TOC anomalies for PJO and nPJO SSWs in WACCM. TOC anomalies are stronger and last longer in the former than in the latter (peak values over the pole around twice as large, 47 versus 25 DU, note the logarithmic scale), indicating that PJO-SSW events have more profound impacts in total column values through deeper alterations of the

stratospheric circulation and associated transport and mixing (see Hitchcock et al., 2013a; de la Cámara et al., 2018, and Figs. 5 and 8). Fusco and Salby (1999) noted the fluctuating nature of TOC, locally sensitive to reversible transport. For instance, the number density increases as air descending along isentropic surfaces compresses, resulting in higher TOC; and this descent must be compensated elsewhere by expansion of air along rising isentropic surfaces. However, reversible transport is unlikely behind the long-lasting, north–south dipole pattern in TOC during the life cycle of SSWs. Figures 4, 5, 6 and 8 all show that

cross-isentropic advection and isentropic irreversible mixing are the main dynamical processes that change ozone mixing ratios during SSWs, in varying proportions at different heights and time lags, and which operate at longer time scales than the driving wave force.

## 4   Summary and conclusions

We have used 240 years of CESM/WACCM climate simulations, run with observed external forcings and boundary conditions

for the period 1955-2014, to quantify variations of Arctic ozone during SSWs and their driving mechanisms. Composites of vertical profiles of polar cap (70°-90°N) anomalies of ozone concentrations on isentropic surfaces during the life cycle of SSWs show common features in MLS data, ERA-Interim, and WACCM (Fig. 3). Starting a few days before the SSW onset, there is more ozone mixing ratio at levels where ozone decreases towards the pole (roughly above 550 K), and less ozone mixing ratio where ozone increases towards the pole (below ∼550 K). Enhanced isentropic eddy transport is the dominant driver of these

anomalies during the onset period. From a zonally-averaged perspective in geographical coordinates, the imbalance between suppressed eddy transport and reinforced cross-isentropic advection is responsible for the slow recovery of the ozone field in the aftermath of the warming that lasts over 1.5 months below ∼600 K (Fig. 4).

There are substantial differences in the timing when the ozone anomalies appear in geographical and equivalent latitude (EqL) averages, which highlight the different dynamical processes involved. In geographical coordinates the initial polar ozone

anomalies grow around one week earlier than in EqL, which indicates the reversible (conservative) nature in the initial changes of the zonal-mean isentropic eddy transport. On the other hand, ozone anomalies in EqL averages ($\phi_e = 70° - 90°$N) appear and disappear at the same time as anomalies in irreversible isentropic mixing of ozone, as diagnosed with the equivalent length $\Lambda_{\mathrm{eq}}^{\mathrm{O3}}$ (Eq. 2) and derived diffusive fluxes (Eq. 3). Particularly in the lower stratosphere, where radiative time scales are much longer than in the upper stratosphere, the gradient-smoothing effect of enhanced isentropic mixing of ozone persists over 2 months

after the SSW onset (Figs. 5 and 6), contributing to the delay of the Arctic ozone recovery in the aftermath of SSWs. The clear temporal offset between enhanced eddy transport of ozone, which operates in the SSW onset, and enhanced irreversible mixing, which operates in the aftermath of the events, is in good agreement with recent estimates of eddy transport and mixing of potential vorticity during SSWs (de la Cámara et al., 2018).



The large sample of SSWs in the WACCM runs (152 in 240 years) allows a statistically robust evaluation of different types of SSW, namely those that are classified as Polar night Jet Oscillation events and those that are not (PJO-SSW and nPJO-SSW, respectively). These two types of SSW are characterized by different evolutions of polar temperature and zonal winds in the aftermath of the SSW (Hitchcock et al., 2013a), and different intensity and duration of the anomalous stratospheric transport

and mixing properties (de la Cámara et al., 2018). We have found that polar ozone undergoes larger variations (anomalies up to 50% as large) that last longer in PJO than in nPJO events Fig. 7. While the evolution of isentropic eddy transport anomalies do not particularly differ between PJO and nPJO SSWs, irreversible isentropic mixing of ozone and mean cross-isentropic advection of ozone (non-conservative effects) are stronger and persist longer in the aftermath of PJO than in nPJO warmings. These are manifestations of larger and more persistent circulation anomalies in the former than in the latter (de la Cámara et al.,

2018; Hitchcock and Shepherd, 2013).

The reported changes in ozone mixing ratios also affect total column values (Fig. 9). TOC estimates from MLS, ERAI, and WACCM present reasonable agreement, with high-latitude increases of ∼47 DU peaking a few days after the SSW onset, and subtropical decreases of around 3.6 DU (MLS column ozone has slightly weaker anomalies). The dipole structure of TOC anomalies lasts around 40-50 days after the SSW onset, but is more persistent in PJO-SSWs (around 2 months) than during

nPJO-SSWs (1 month).

The results of the present study contribute to a better interpretation of the observed interannual variability of Arctic ozone and a better quantification of its evolution, with particular emphasis on the effects of irreversible mixing. However, the impacts of SSWs on the ozone field reach tropical latitudes as suggested in Fig. 9. The exploration of tropical ozone variability during SSWs, and its interaction with the Quasi Biennial Oscillation, will be explored in a future study.

*Data availability.* Data from ERA-Interim are freely available at http://apps.ecmwf.int/datasets/. The output from the WACCM runs is available at https://www2.acom.ucar.edu/gcm/ccmi-output, and also upon request to the corresponding author. The SWOOSH dataset is available at https://www.esrl.noaa.gov/csd/groups/csd8/swoosh/.

*Author contributions.* AdlC, MA and PH conceived and designed the study. AdlC performed the data analysis with input from all co-authors on the interpretation and discussion of results. AdlC led the manuscript writing with significant contributions from all co-authors.

*Competing interests.* The authors declare that they have no conflict of interest

*Acknowledgements.* This work has been partially funded by Spanish national projects PALEOSTRAT (CGL2015-69699) and STEADY (CGL2017-83198-R). M. Abalos has been supported by the research grant Atracción de Talento Comunidad de Madrid (ref: 2016-T2/AMB-1405).





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





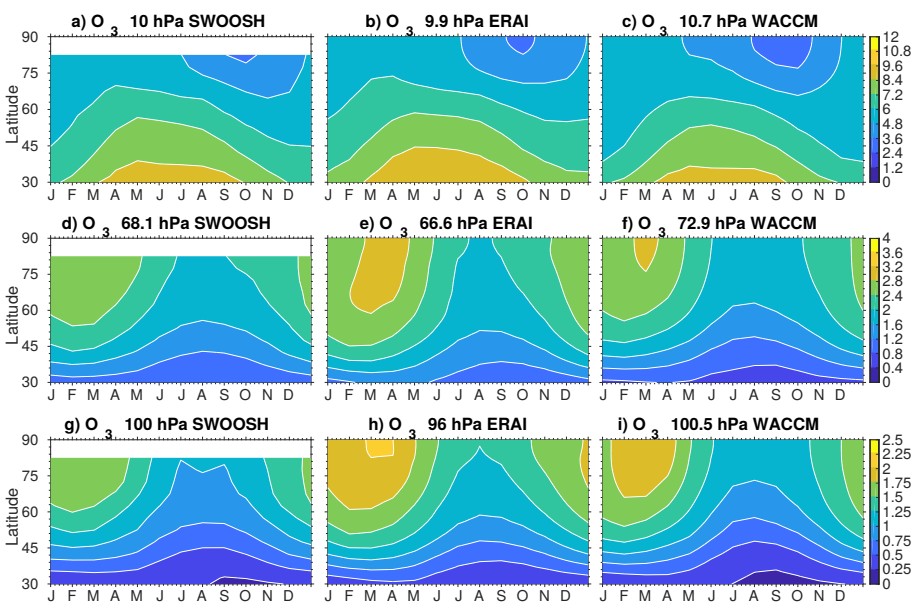

**Figure 1.** Climatological seasonal cycle of zonal-mean ozone (ppmv) north of 30°N at different pressure surfaces in the lower and mid-stratosphere for (left column) SWOOSH data, (central column) ERA-Interim, and (right column) WACCM output.





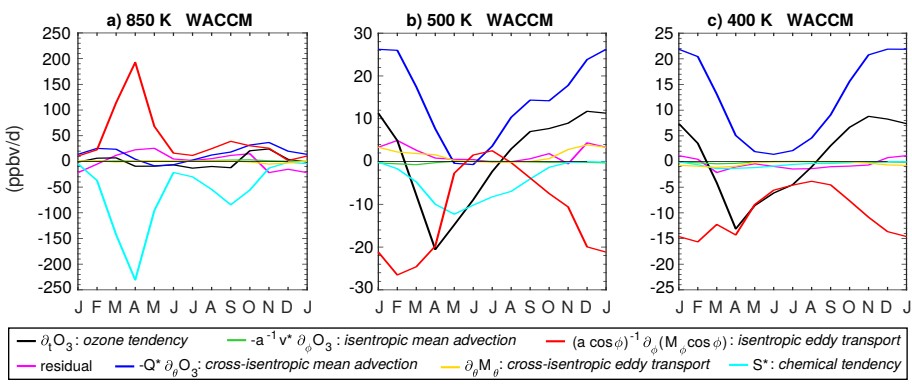

**Figure 2.** Climatological seasonal cycle of the different terms in the ozone continuity equation (Eq. 1) averaged over the Arctic (70°-90°N) at a) 850 K, b) 500 K and c) 400 K. WACCM output.





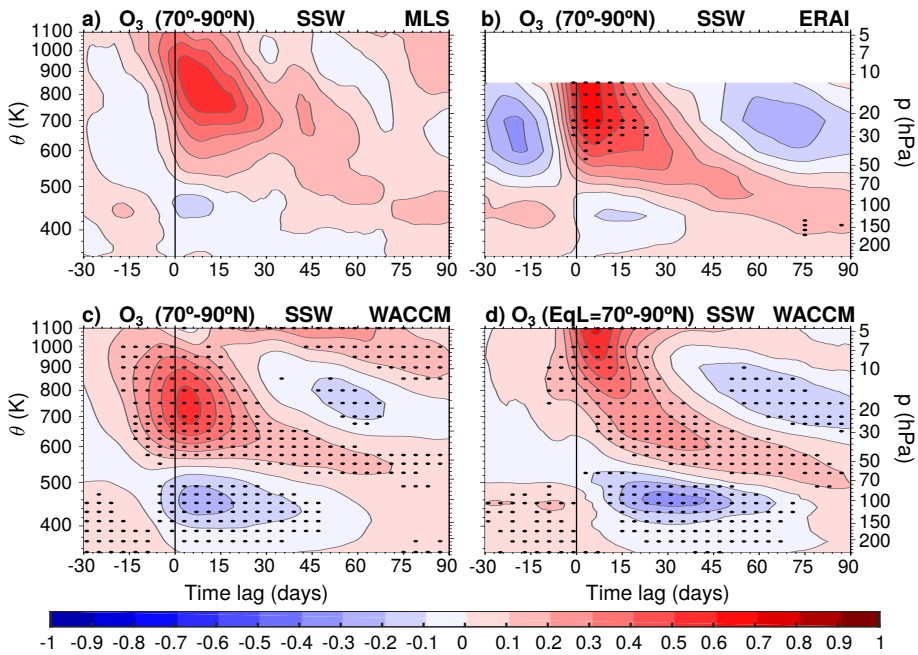

**Figure 3.** Composite evolution around the SSW central day as a function of potential temperature of ozone concentration anomalies (ppmv) averaged over $70°$-$90°$N for a) MLS, b) ERAI, and c) WACCM; d) similar composite but averaging over EqL $70°$-$90°$N in WACCM. Black dots denote statistically significant anomalies (two-tailed Student t-test, $\alpha = 0.01$). Note that the statistical test has not been performed for MLS due to the small sample of SSWs. The approximate pressure level is indicated in the right axis.





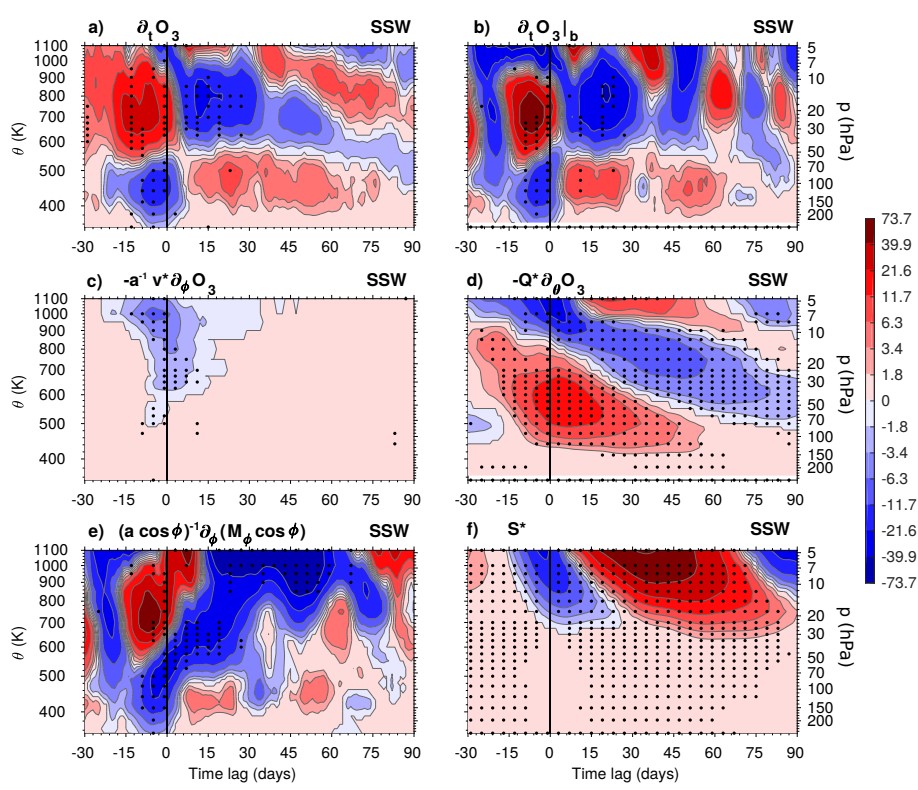

**Figure 4.** Composite evolution, centered on the SSW central date, of the anomalies of the different terms in the zonal-mean ozone continuity equation (Eq. 1) (in ppbv/d) as a function of potential temperature, averaged over $70°$-$90°$N, for WACCM. Black dots indicate statistically significant values (two-tailed Student t-test, $\alpha = 0.01$).





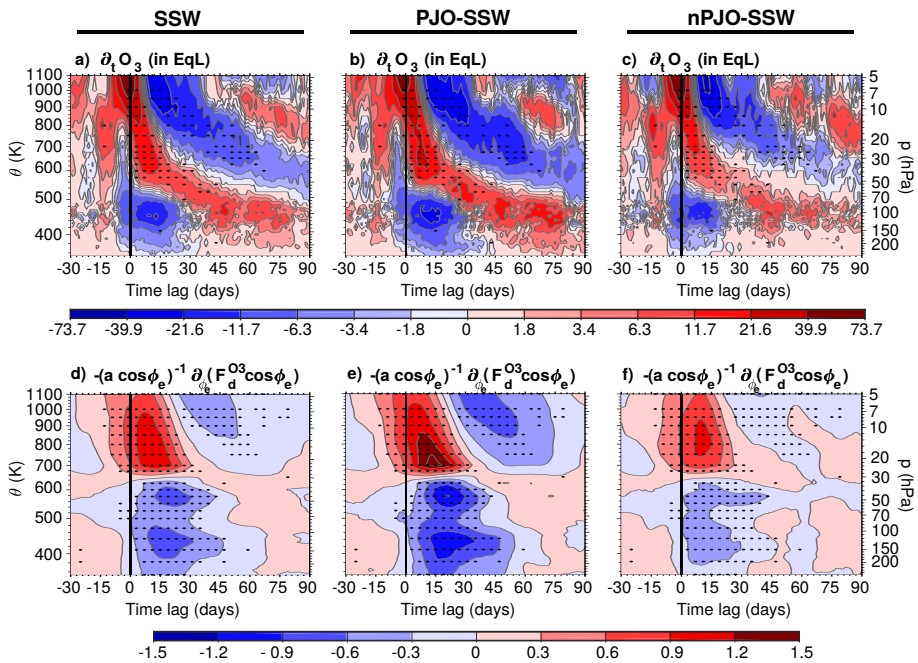

**Figure 5.** Composite evolution, centered on the SSW central date and averaged over equivalent latitudes $\phi_e = 70°$-$90°$N, of (top panels) anomalies of ozone tendencies (ppbv/d), and (bottom panels) the standardized anomalies of mixing-induced ozone tendencies (see Eq. 3). (Left column) All SSW events in WACCM, (middle column) PJO-SSW events, and (right column) nPJO-SSW events. Black dots indicate statistically significant values (two-tailed Student t-test, $\alpha = 0.01$).





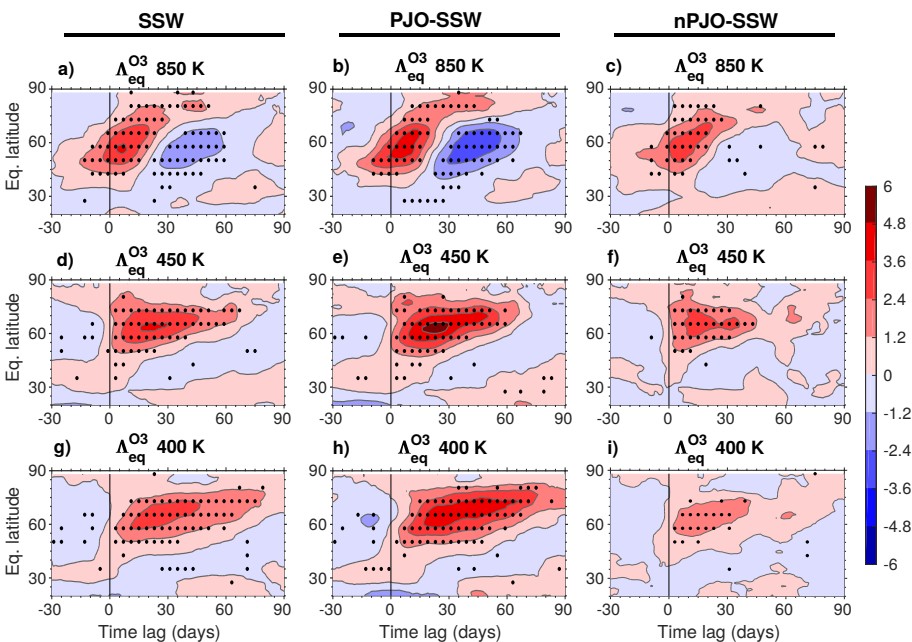

**Figure 6.** Composite evolution centered on the SSW central date as a function of EqL of equivalent length $\Lambda_{eq}^{O3}$ anomalies (non-dimensional units), for (left column) all SSW events, (central column) PJO-SSW events, and (right column) nPJO-SSW events in WACCM, at 850 K, 500 K and 400 K as indicated. Black dots indicate statistically significant values (two-tailed Student t-test, $\alpha = 0.01$).





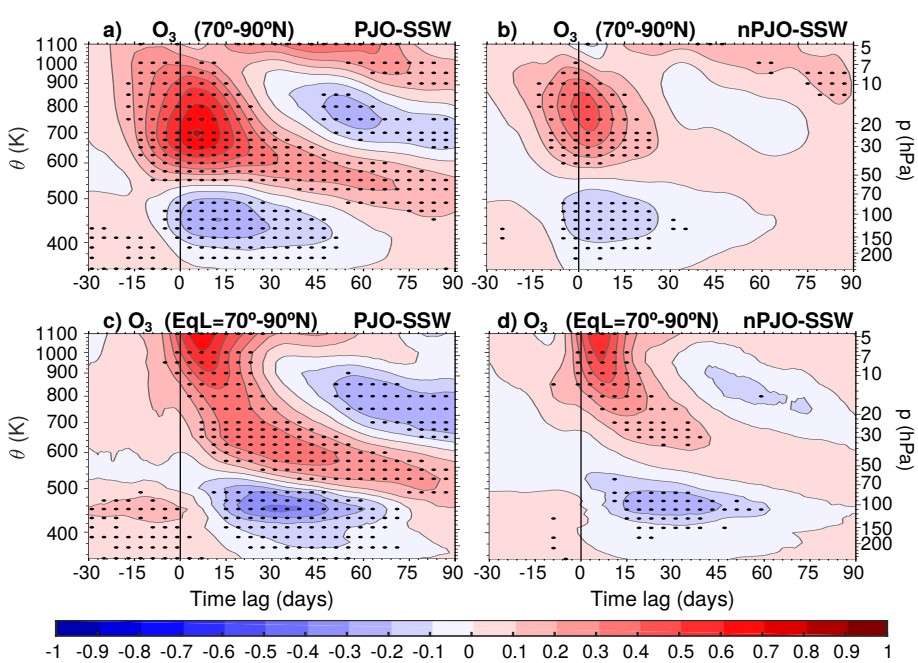

**Figure 7.** As in Fig. 3, but for PJO-SSW and nPJO-SSW events in WACCM.



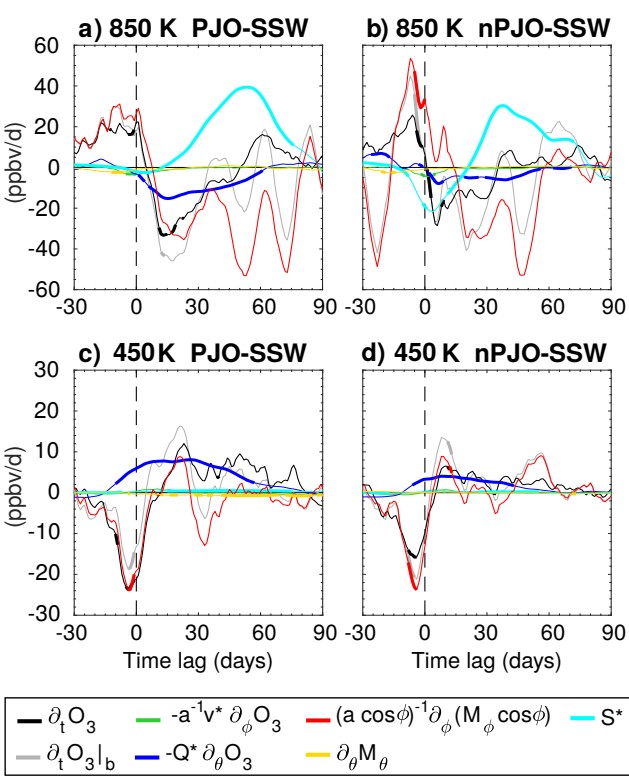

**Figure 8.** Composite evolution of the anomalies of the different terms in Eq. 1 at 850 and 450 K, averaged over $70°$-$90°$N. (Left column) PJO-SSW, and (right column) nPJO-SSW. Thick lines indicate statistically significant values (two-tailed Student t-test, $\alpha = 0.01$). WACCM output.





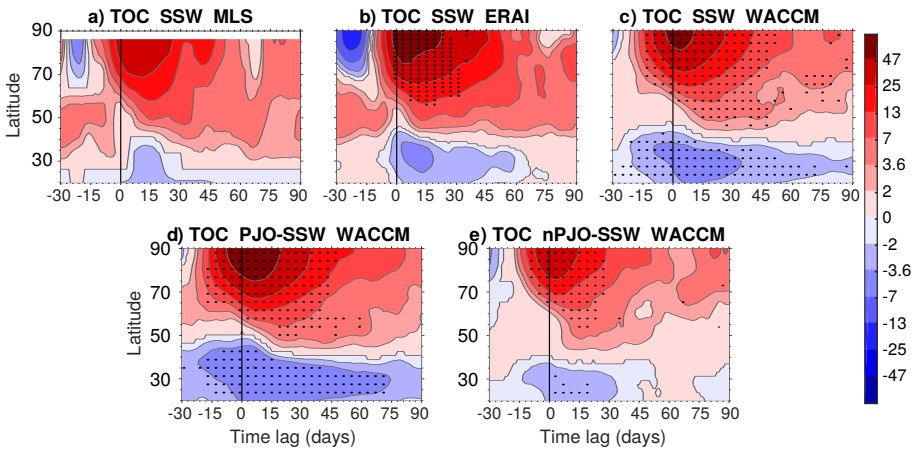

**Figure 9.** Composite evolution, centered on the SSW central date, of total ozone column (TOC) anomalies (in DU) as a function of latitude. Composite for a) SSWs in MLS, b) SSWs in ERAI, c) SSWs in WACCM, d) PJO-SSWs in WACCM, and e) nPJO-SSW in WACCM. Black dotes indicate statistically significant values (two-tailed Student t-test, $\alpha = 0.01$) (the statistical test has not been performed for MLS data due to the small sample of SSWs).