# Peer review of "Response of Arctic ozone to sudden stratospheric warmings"

_Atmospheric Chemistry and Physics, 2018_

## Referee Comment (RC1) · Anonymous Referee #1 · 30 Aug 2018

TITLE: Response of Arctic ozone to sudden stratospheric warmings

AUTHORS: de la Camara, A., Abalos, M., Hitchcock, P., Calvo, N., and Garcia, R. R.

Summary

This paper examines the polar ozone budget during major stratospheric sudden warming (SSW) events based on 240 NH winters generated by the WACCM global model system. When appropriate WACCM ozone results are compared with observations (MLS), a combined observation data set (SWOOSH), and with a reanalysis (ERA-Interm). The goal is to better understanding the evolution, vertical structure, and the specific dynamical processes that control the significant polar ozone changes occurring during SSW events. As part of this work the role of the vertical depth of the warming,

classified as PJO or non-PJO (nPJO) is examined specifically along with changes in total column ozone distribution. By identifying and compositing the WACCM generated SSW events, composite changes prior to and following the SSW can be examined in detail. Results show that the PJO SSW events are stronger in the growth phase and more persistent in the decay phase than the nPJO events since the PJO SSW events descend to the lowest part of the stratosphere (stronger events) where radiative time scales are longer (persistent decay).

Strengths:

Overall, this is a well written paper with clear and appropriate figures. Detailed understanding of ozone variability and budgets continues to be relevant scientific question well within the scope of ACP. This research is based on a state-of-the-art global model especially well suited to ozone budget studies and makes use of two approaches to the ozone budget: one based on equivalent latitude coordinates and a second, more conventional but detailed, Eulerian budget. The paper clearly lays out these potentially confusing budget terms and their physical interpretation. The methods and assumptions are valid and clearly presented and the results support the substantial conclusions. The methods are presented in enough detail that they can be duplicated by other global models, though there could be some interesting differences between model results if a different model is used. The title is appropriate, the abstract is concise and complete, and the reference list is thorough.

Weaknesses:

There are no weaknesses as such but a comment is made below.

Comment:

Some of the conclusions might benefit from WACCM qualification as most of the conclusions are derived from the WACCM diagnostics, especially as the WACCM model was shown to differ from the ERAI reanalysis in the pre SSW time. Specifically:

Page 9, Line 7: "This is fast, occurs in the week or two prior to the central dates of the composite, and is reversible (e.g. if the wave packet propagates through and the contours return to zonal)."

and Page 11, Line 23: "There are substantial differences in the timing when the ozone anomalies appear in geographical and equivalent latitude (EqL) averages, which highlight the different dynamical processes involved.

These statements seem to apply mainly to the WACCM results in the period prior to the SSW events, while, in Fig. 3, ERAI lacks significant changes a week or two prior to the SSW events, so it appears still open as to how much these statements apply more generally.

The context is usual clear, however, maybe adding just a WACCM qualifying sentence to the Discussion and Conclusion section would be useful to readers.

Minor Remarks:

1) Page 2, Line 1: "PLUMB"

2) Page 2, Line 10: "short" (shortly)

3) Page 4, Line 31: Reference could be added: Butchart, N. and E.E. Remsberg, 1986: The Area of the Stratospheric Polar Vortex as a Diagnostic for Tracer Transport on an Isentropic Surface. J. Atmos. Sci., 43, 1319–1339, https://doi.org/10.1175/1520-0469(1986)043<1319:TAOTSP>2.0.CO;2

4) Page 7, Line 2: Note that Fig. 3d will be discussed later.

5) page 7, Lines 15-22: The differences between the two ozone tendencies are intriguing. Could another source of the discrepancy be numerical diffusion? Parameterized GWD diffusion might be too weak in the stratosphere to account for the difference. Another possibility is just the different formulation and differencing schemes between the model and the off-line calculated diagnostics.

6) page 7, Line 33: Maybe after "ozone mixing ratio" add a reference to "(Fig. 3c)".

---

## Referee Comment (RC2) · M. Tao (Referee) · 22 Sep 2018

General comments

The paper analyzed the polar ozone evolutions during the SSWs using the WACCM 240-year simulation, which is also shown a fair agreement with the ERA-Interim, MLS and SWOOSH ozone. Then, the contribution from relevant dynamical (e.g. isentropic mean advection, isentropic eddy transport, cross-isentropic advective transport, cross-isentropic eddy trasnport) and chemical processes are quantitively diagnosed.

One highlight of this work is to provide statistical analysis of ozone changed during PJO-SSW and nPJO-SSW based on the long-term WACCM run. The result shows that the polar ozone anomalies are stronger and longer during PJO-SSW than nPJO-

[Figure]

SSW, which is found related to the irreversible mixing and cross-isentropic advection.

Another highlight is that the authors particularly used equivalent length for ozone to quantify the irreversible mixing and make a note of the difference between the irreversible mixing and eddy transport. The differences between the 'Lagrangian' and 'Eulerian' perspective deepen the understanding of the roles of these processes.

The paper is well-written with valid methods, a clear structure and appropriate figures. The topic of the study meets the scope of ACP well. The flow of the whole paper is very clear and brief. The conclusions are sufficiently supported by the materials. In general, this paper has a good scientific quality and I recommend it to be published on ACP. I only have a few suggestions to authors, which can potentially improve the presentation of the paper.

Specific comments

1. Page 5, line15-20: please specified which temperature and wind are used for the SSW identification for MLS and SWOOSH.

2. The statements about Figure 1: better to mention the climatological seasonality is based on different length of climatology, i.e. 1980-2017 for SWOOSH, 2004-2012 for MLS, 1979-2012 for ERA-I and 240-year for WACCM, either in the text or in the caption of the figure.

3. Page 6, line 8 and line 21, please check the writing of the citation: Brasseur, Guy P. and Susan Solomon.ÂăAlso that in the reference list.

4. A suggestion for Fig. 4 and 8: to add the physical interpretations for each mathematics terms are helpful to readers. I would add the physical interpretations like what is done in Fig.2 also in this two figures.

5. Also a suggestion to easier go back and forward from text to figure: add some brackets with the color of the corresponding lines after the physical interpretations in the last paragraph of page 9, e.g. 'isentropic eddy transport (red line)' or 'vertical

advection of ozone (dark blue line)'.

---

## Author Comment (AC1) · 22 Oct 2018

**Response to Reviewer #1**

**Reviewers' comments are in** *italics***; our responses are in bold.**

**The references to specific manuscript lines in our responses refer to manuscript lines in the track-change version.**

*Summary*
*This paper examines the polar ozone budget during major stratospheric sudden warming (SSW) events based on 240 NH winters generated by the WACCM global model system. When appropriate WACCM ozone results are compared with observations (MLS), a combined observation data set (SWOOSH), and with a reanalysis (ERA-Interm). The goal is to better understanding the evolution, vertical structure, and the specific dynamical processes that control the significant polar ozone changes occurring during SSW events. As part of this work the role of the vertical depth of the warming,classified as PJO or non-PJO (nPJO) is examined specifically along with changes in total column ozone distribution. By identifying and compositing the WACCM generated SSW events, composite changes prior to and following the SSW can be examined in detail. Results show that the PJO SSW events are stronger in the growth phase and more persistent in the decay phase than the nPJO events since the PJO SSW events descend to the lowest part of the stratosphere (stronger events) where radiative time scales are longer (persistent decay).*

*Strengths:*
*Overall, this is a well written paper with clear and appropriate figures. Detailed understanding of ozone variability and budgets continues to be relevant scientific question well within the scope of ACP. This research is based on a state-of-the-art global model especially well suited to ozone budget studies and makes use of two approaches to the ozone budget: one based on equivalent latitude coordinates and a second, more conventional but detailed, Eulerian budget. The paper clearly lays out these potentially confusing budget terms and their physical interpretation. The methods and assumptions are valid and clearly presented and the results support the substantial conclusions. The methods are presented in enough detail that they can be duplicated by other global models, though there could be some interesting differences between model results if a different model is used. The title is appropriate, the abstract is concise and complete, and the reference list is thorough.*

*Weaknesses:*
*There are no weaknesses as such but a comment is made below.*

**We thank the reviewer for his/her positive assessment. We have followed the suggestions and changed the text accordingly; our point-by-point responses are below.**

*Comment:*
*Some of the conclusions might benefit from WACCM qualification as most of the conclusions are derived from the WACCM diagnostics, especially as the WACCM model was shown to differ from the ERAI reanalysis in the pre SSW time. Specifically:*
*Page 9, Line 7: "This is fast, occurs in the week or two prior to the central dates of the composite, and is reversible (e.g. if the wave packet propagates through and the contours return to zonal)."*
*and Page 11, Line 23: "There are substantial differences in the timing when the ozone anomalies appear in geographical and equivalent latitude (EqL) averages, which highlight the different dynamical processes involved.*

*These statements seem to apply mainly to the WACCM results in the period prior to the SSW events, while, in Fig. 3, ERAI lacks significant changes a week or two prior to the SSW events, so it appears still open as to how much these statements apply more generally.*

*The context is usual clear, however, maybe adding just a WACCM qualifying sentence to the Discussion and Conclusion section would be useful to readers.*

**We agree with the reviewer that our conclusions on the different dynamical processes involved in the evolution of ozone during SSW are drawn from WACCM output. But the initial changes in ozone during the pre-SSW period (lags -15 to 0) are present both in WACCM and ERAI. The main difference between Fig. 3b and 3c (ozone composite for ERAI and WACCM) is that WACCM does not show the negative anomalies in the pre-warming period above 500K. On the other hand, the ozone tendency is positive starting at lag -15 both in ERAI and WACCM. So we have reasons to think that the description of the dynamics using WACCM is valid to understand the ozone evolution in ERAI as well. We have clarified this point in page 9 lines 15-18, and page 11 line 30.**

*Minor Remarks:*

*1) Page 2, Line 1: "PLUMB"*

**Thank you, we have revised this and other references in our manuscript.**

*2) Page 2, Line 10: "short" (shortly)*

**Corrected (page 2 line 9).**

*3) Page 4, Line 31: Reference could be added: Butchart, N. and E.E. Remsberg, 1986: The Area of the Stratospheric Polar Vortex as a Diagnostic for Tracer Transport on an Isentropic Surface. J. Atmos. Sci., 43, 1319–1339, https://doi.org/10.1175/1520-0469(1986)043<1319:TAOTSP>2.0.CO;2*

**This is a relevant reference, we have included it in page 4 line 30.**

**We have also included a reference to Lubis et al. (2017 ACP) in the first paragraph of the introduction.**

*4) Page 7, Line 2: Note that Fig. 3d will be discussed later.*

**We have introduced this clarification in page 7 line 3.**

*5) page 7, Lines 15-22: The differences between the two ozone tendencies are intriguing. Could another source of the discrepancy be numerical diffusion? Parameterized GWD diffusion might be too weak in the stratosphere to account for the difference. Another possibility is just the different formulation and differencing schemes between the model and the off-line calculated diagnostics.*

**This is a good point. Indeed, we agree with the reviewer that numerical diffusion and specially the different numerical formulation between the model's transport scheme and our off-line diagnostics (Eq. 1) are important sources of discrepancy. We have added a sentence in page 7 lines 20-24.**

*6) page 7, Line 33: Maybe after "ozone mixing ratio" add a reference to "(Fig. 3c)".*

**Done (page 8 line 1).**

---

## Author Comment (AC2) · 22 Oct 2018

**Response to Reviewer #2 (Mengchu Tao)**

**Reviewers' comments are in** *italics*; **our responses are in bold.The references to specific manuscript lines in our responses refer to manuscript lines in the track-change version.**

*General comments*
*The paper analyzed the polar ozone evolutions during the SSWs using the WACCM 240-year simulation, which is also shown a fair agreement with the ERA-Interim, MLS and SWOOSH ozone. Then, the contribution from relevant dynamical (e.g. isentropic mean advection, isentropic eddy transport, cross-isentropic advective transport, cross-isentropic eddy trasnport) and chemical processes are quantitively diagnosed. One highlight of this work is to provide statistical analysis of ozone changed during PJO-SSW and nPJO-SSW based on the long-term WACCM run. The result shows that the polar ozone anomalies are stronger and longer during PJO-SSW than nPJO-SSW, which is found related to the irreversible mixing and cross-isentropic advection. Another highlight is that the authors particularly used equivalent length for ozone to quantify the irreversible mixing and make a note of the difference between the irreversible mixing and eddy transport. The differences between the 'Lagrangian' and 'Eulerian' perspective deepen the understanding of the roles of these processes.*
*The paper is well-written with valid methods, a clear structure and appropriate figures. The topic of the study meets the scope of ACP well. The flow of the whole paper is very clear and brief. The conclusions are sufficiently supported by the materials. In general, this paper has a good scientific quality and I recommend it to be published on ACP. I only have a few suggestions to authors, which can potentially improve the presentation of the paper.*
**We thank Mengchu Tao for her positive assessment. We have followed her suggestions and changed the text accordingly; our point-by-point responses are below.**

*Specific comments*
*1. Page 5, line15-20: please specified which temperature and wind are used for the SSW identification for MLS and SWOOSH.*
**We have used the SSW dates identified with ERA-Interim to composite the MLS ozone data. We have added a sentence clarifying this point in page 5 line 21.**
**Note that SWOOSH provides monthly averages, and we have just used this dataset to look at the climatological seasonal cycle in Fig. 1.**

*2. The statements about Figure 1: better to mention the climatological seasonality is based on different length of climatology, i.e. 1980-2017 for SWOOSH, 2004-2012 for MLS, 1979-2012 for ERA-I and 240-year for WACCM, either in the text or in the caption of the figure.*
**We agree, we have done as suggested, see page 6 lines 5-6.**

*3. Page 6, line 8 and line 21, please check the writing of the citation: Brasseur, Guy P. and Susan Solomon. Also that in the reference list.*
**Thank you, we have revised this and other references.**
**We have also included a reference to Lubis et al. (2017 ACP) in the first paragraph of the introduction.**

*4. A suggestion for Fig. 4 and 8: to add the physical interpretations for each mathematics terms are helpful to readers. I would add the physical interpretations like what is done in Fig.2 also in this two figures.*
**We agree that including the physical interpretation of each term of the continuity equation makes things easier to the reader. We have modified legends/titles in Figs. 4 and 8 accordingly.**

*5. Also a suggestion to easier go back and forward from text to figure: add some brackets with the color of the corresponding lines after the physical interpretations in the last paragraph of page 9, e.g. 'isentropic eddy transport (red line)' or 'vertical advection of ozone (dark blue line)'.*
**We have followed this suggestion in the discussion of Fig. 8 (paragraph starting in page 9 line 27).**